# *LDOC1* Suppresses Microbe-Induced Production of IL-1β in Human Normal and Cancerous Oral Cells through the PI3K/Akt/GSK-3β Axis

**DOI:** 10.3390/cancers12113148

**Published:** 2020-10-27

**Authors:** Chia-Huei Lee, Pin-Feng Hung, Ko-Jiunn Liu, Hsuan-Lien Chung, Wen-Chan Yang, Kai-Cheng Hsu, Tsorng-Harn Fong, Hsiu-Jung Lo, Ya-Ping Chen, Ji-Rui Yang, Ching-Yu Yen

**Affiliations:** 1National Institute of Cancer Research, National Health Research Institutes, 35 Keyan Road, Zhunan 35053, Taiwan; hdp91111@nhri.edu.tw (P.-F.H.); kojiunn@nhri.edu.tw (K.-J.L); zzz810355@nhri.edu.tw (H.-L.C.); 020801@nhri.edu.tw (W.-C.Y.); 090406@nhri.edu.tw (Y.-P.C.); jry7921@gmail.com (J.-R.Y.); 2Department of Oral and Maxillofacial Surgery, Chi Mei Medical Center, Tainan 710, Taiwan; shining198715@gmail.com; 3Department of Anatomy and Cell Biology, School of Medicine, College of Medicine, Taipei Medical University, Taipei 110, Taiwan; thfong@tmu.edu.tw; 4National Institute of Infectious Disease and Vaccinology, National Health Research Institutes, Zhunan 350, Taiwan; hjlo@nhri.edu.tw; 5School of Dentistry, China Medical University, Taichung 404, Taiwan; 6School of Dentistry, Taipei Medical University, Taipei 110, Taiwan

**Keywords:** poor oral hygiene (POH), leucine-zipper downregulated in cancer 1 (*LDOC1*), *Candida albicans*, interleukin 1beta (IL-1β), oral squamous cell carcinoma (OSCC), Protein Kinase B (PKB *Akt)*, Phosphoinositide 3-kinase (PI3K), Glycogen Synthase Kinase-3 Beta (GSK-3β)

## Abstract

**Simple Summary:**

Oral microbes often proliferate due to poor oral hygiene (POH). POH is associated with OSCC (oral squamous cell carcinoma). We investigated the role of *LDOC1* in the production of IL-1β, an oncogenic proinflammatory cytokine in OSCC, induced by microorganisms in human oral cells. *Candida albicans* (*CA*) was detected in OSCC tissues. *CA* and the oral bacterium *Fusobacterium nucleatum* stimulate higher levels of IL-1β production in *LDOC1*-deficient OSCC cells than in *LDOC1*-expressing oral cells. *CA* SC5314 increased OSCC incidence in carcinogen-treated mice. Loss and gain of *LDOC1* function resulted in increased and decreased, respectively, *CA* SC5314-induced IL-1β production. *LDOC1* deficiency increased active pAkt^S473^ upon SC5314 stimulation and inactive pGSK-3β^S9^ phosphorylated by pAkt^S473^. PI3K and Akt inhibitors and expression of constitutively active mutant GSK-3β^S9A^ reduced the SC5314-stimulated IL-1β production in *LDOC1*-deficient cells. These results indicate that the PI3K/Akt/pGSK-3β signaling contributes to *LDOC1*-mediated inhibition of microbe-induced IL-1β production, suggesting *LDOC1* may determine the role of oral microbes in POH-associated OSCC.

**Abstract:**

Poor oral hygiene (POH) is associated with oral squamous cell carcinoma (OSCC). Oral microbes often proliferate due to POH. Array data show that *LDOC1* plays a role in immunity against pathogens. We investigated whether *LDOC1* regulates the production of oral microbe-induced IL-1β, an oncogenic proinflammatory cytokine in OSCC. We demonstrated the presence of *Candida albicans* (*CA*) in 11.3% of OSCC tissues (*n* = 80). *CA* and the oral bacterium *Fusobacterium nucleatum* stimulate higher levels of IL-1β secretion by *LDOC1*-deficient OSCC cells than by *LDOC1*-expressing oral cells. *CA* SC5314 increased OSCC incidence in 4-NQO (a synthetic tobacco carcinogen) and arecoline-cotreated mice. Loss and gain of *LDOC1* function significantly increased and decreased, respectively, *CA* SC5314-induced IL-1β production in oral and OSCC cell lines. Mechanistic studies showed that *LDOC1* deficiency increased active phosphorylated Akt upon *CA* SC5314 stimulation and subsequent inhibitory phosphorylation of GSK-3β^S9^ by activated Akt. PI3K and Akt inhibitors and expression of the constitutively active mutant GSK-3β^S9A^ significantly reduced the *CA* SC5314-stimulated IL-1β production in *LDOC1*-deficient cells. These results indicate that the PI3K/Akt/pGSK-3β signaling pathway contributes to *LDOC1*-mediated inhibition of oral microbe-induced IL-1β production, suggesting that *LDOC1* may determine the pathogenic role of oral microbes in POH-associated OSCC.

## 1. Introduction

Approximately 350,000 new cases of oral cancer are diagnosed annually worldwide [1]. Oral squamous cell carcinomas (OSCCs) account for over 90% of oral cancers [2,3]. The risk of OSCC is substantially increased by alcohol consumption, betel quid chewing, and cigarette smoking (collectively termed “ABC”) [4,5]. However, most consumers of alcohol, betel quid, and tobacco will not develop OSCC in their lifetime, indicating that other risk factors are involved. The American Society of Clinical Oncology stated that factors other than ABC, including gender, poor oral hygiene (POH), and a weak immune system, can increase a person’s risk of developing oral cancer [6]. Personal oral hygiene and immunity have profound impact on oral microbiota, which may be an important factor in the OSCC tumor microenvironment. The relationship between oral microbes and OSCC has been noted [7,8,9]. Over 700 species of microorganisms have been identified in the oral cavity of humans [10]. Dysbiosis of oral microbial flora induces oral inflammation and contributes to numerous human diseases [11], including OSCC. Oral microbial imbalance dominated by *Candida albicans* (*CA*) was identified in OSCC patients [7,8,9]. Analysis of oral microbiota performed in healthy control subjects and patients with OSCC revealed that the species richness and diversity were significantly lower in patients with OSCC. *CA* was the most abundant microorganisms identified in the oral cavities of patients with OSCC [8]. *CA* can cause various oral mucosal lesions, including chronic hyperplastic candidiasis (also known as candidal leukoplakia) [9,12], which may increase the risk of OSCC. Furthermore, *CA* may promote oral malignant transformation from precancer lesions, such as leukoplakia [13]. *CA* is recognized as an unharmful commensal fungal species in healthy individuals. It can, however, become pathogenic in an immunocompromised person. Consistently, several studies have reported the development of oral carcinoma in immunocompromised patients with chronic mucocutaneous candidiasis [14,15,16]. Besides *CA*, an epidemiologic study identified that three species of periodontal bacteria, *Prevotella tannerae*, *Fusobacterium nucleatum* (*FN*), and *Prevotella intermedia*, had a positive association with OSCC risk, and this relationship may be influenced by lifestyle and genetic factors [17]. The oncogenic role of *FN* has been noted in colorectal cancer [18,19]. Previous investigation showed that *FN* can induce immune suppression in gut mucosa by suppressing the functions of immune cells [18]. Although the association between *CA* and *FN* and OSCC has been noted, it is not clear whether these microorganisms are protumor factors of POH-associated OSCC, and what may be involved in oral microbial carcinogenesis.

Numerous studies have investigated salivary transcriptome and proteome to identify molecules with diagnostic value for OSCC. The results have revealed the potential of specific deregulated cytokines, such as IL-1β, IL-6, and IL-8, as predictive salivary biomarkers for the early diagnosis of oral cancer [20,21,22,23]. Not only a OSCC biomarker, IL-1β also promotes the development and progression of OSCC in cell-based assays and mouse models by substantially inducing IL-6 and IL-8 expression to create a pro-oncogenic microenvironment through autocrine signaling [24]. Moreover, IL-1β transactivates the epidermal growth factor receptor through the CXCL1–CXCR2 axis in oral cancer [25]. These results indicate the protumor activities of IL-1β in OSCC. However, the causal factors that induce elevated IL-1β in OSCC remained unidentified. There is a possibility that microbes in the oral cavity, such as *CA* and *FN*, may substantially increase IL-1β levels in immunocompromised individuals and promote the development and progression of OSCC.

Our previous study identified leucine-zipper downregulated in cancer 1 (*LDOC1*), as an X-linked tumor suppressor in OSCC. *LDOC1* was frequently silenced by promoter hypermethylation in OSCC patients who habitually smoke cigarettes [26]. Upon analysis of LDOC1 protein expression in an immortalized normal oral keratinocyte cell line CGHNK2 [27], a dysplasia oral keratinocyte (DOK) cell line derived from a heavy smoker [28], and four OSCC cell lines established from patients who were smokers [29], we observed high expression of *LDOC1* in CGHNK2 cells, whereas it was downregulated in the DOK cell line and nearly undetectable in all OSCC cell lines tested (Appendix A). Our previous studies also demonstrated that the promoter methylation of *LDOC1* is tobacco sensitive. *LDOC1* was silenced by promoter hypermethylation after CGHNK2 cells were exposed to cigarette smoke condensates (CSCs) for only 6 weeks [30]. *LDOC1* was also epigenetically silenced by promoter hypermethylation in human bronchial epithelial BEAS-2B cells by treatment with CSCs [31]. The *LDOC1* gene encodes a protein of 146 amino acids, which displays a typical leucine-zipper motif in the N-terminal domain and a proline-rich region that is similar to SH3-binding domains [32]. To explore the major function of *LDOC1* in human oral cells, we generated an *LDOC1* knockdown cell line and the corresponding control derived from CGHNK2 cells (CGHNK2-shLDOC1 and CGHNK2-shCtrl, respectively) (Appendix A). The results from cDNA microarray (GSE149108) and Ingenuity Pathway Analysis revealed that four out of the top five dysregulated canonical pathways caused by *LDOC1* knockdown in the CGHNK2 cells were more relevant to immune response (Appendix A). Networks with functions associated with antimicrobial and inflammatory responses were the most upregulated networks in CGHNK2-shLDOC1 cells (Appendix A). These preliminarily data suggested that *LDOC1* may be involved in the immune response to microbial infection. Accumulating evidence indicates that habitual cigarette smoking can alter a person’s immune response through epigenetic mechanisms [20,33]. The sensitivity of *LDOC1* promoter methylation to cigarettes thus raised the possibility that loss of *LDOC1* function may be a reason for immune abnormalities in smokers. In the setting of *LDOC1* silencing, oral microorganisms such as *CA* and *FN* may become pro-tumor through stimulating production of oncogenic IL-1β.

In the present study, we examined the presence of *CA* in OSCC samples. We assessed the impact of *CA* on oral carcinogenesis in mice. The effects of *LDOC1* on *CA*- and *FN*-induced production of IL-1β in oral normal CGHNK2 and OSCC TW2.6 cell lines were investigated. Mechanistic studies were also conducted to investigate the molecular network through which *LDOC1* regulates IL-1β production upon oral microorganism stimulation.

## 2. Results

### 2.1. CA Is Present in OSCC Tumors and Non-Tumor Tissues and Invades Human Oral Keratinocytes and OSCC Cells

A dysbiotic mycobiome dominated by *CA* was identified in OSCC [8]. The Gomori methenamine silver (GMS) method of staining was used to detect fungi and confirm the presence of *CA* in the oral cavity of patients with OSCC. We determined that 13.8% (11/80) of OSCC cases were GMS-positive (black stained) and infected by fungi (Figure 1a, left panels). Representative samples that were negative for GMS staining are presented in the right panels of Figure 1a. No differences were observed in fungi colonization status between tumor and peritumoral tissues in the same patient. We performed *CA*-specific real-time PCR with genomic DNA isolated from all GMS-positive and 30 randomly selected GMS-negative samples to examine whether *CA* was present in the GMS-positive specimens. The PCR results indicated that *CA* was present in 9 of 11 GMS-positive samples, accounting for a 11.3% (9/80) infection rate, whereas it was not observed in any of the GMS-negative samples tested. Representative gel views of PCR amplicon fragments are displayed in Figure 1b. The invasion of the *CA* strain SC5314 expressing GFP (SC5314-GFP) in an immortalized normal oral keratinocyte CGHNK2 cell line and an OSCC TW2.6 cell line was confirmed by immunofluorescence confocal imaging, which revealed intracellular SC5314-GFP (green) 1 h after the coculture (Figure 1c). These results demonstrated that *CA* is present in OSCC samples and invades human normal and cancerous oral cells.

### 2.2. Oral Microbes May Contribute to High Levels of IL-1β in OSCC with LDOC1 Downregulation

We then examined whether *CA* and *FN* can stimulate human oral cells to secrete IL-1β. After cells were cocultured with preparations of either live (L) or heat-killed (HK) *CA* SC5314 (MOI of 0.5) for 24 h, IL-1β levels in the conditioned medium of all cells tested were significantly elevated, with *LDOC1*-deficient TW2.6 cells exhibiting a larger increase compared with *LDOC1*-expressing CGHNK2 cells. In both cell lines tested, L *CA* more strongly induced IL-1β than HK *CA* (Figure 2a). In addition to *CA*, live *FN* (OD_600_ = 0.08) can also stimulate CGHNK2 and TW2.6 cell lines to secrete IL-1β during coculture for 6 h (Figure 2b). Similarly, the IL-1β levels induced by *FN* were significantly higher in *LDOC1*-deficient TW2.6 than those of *LDOC1*-expressing CGHNK2 (Figure 2b). These results suggested that the presence of oral microbes, such as *CA* and *FN*, may contribute to high levels of IL-1β in OSCC tumors, especially in OSCC cells with *LDOC1* downregulation. To assess the correlation between *LDOC1* and IL-1β, we explored the mRNA expression of *LDOC1* and IL-1β in OSCC specimens by surveying public microarray data deposited on the Oncomine website. In Peng’s study [34], mRNA expression of *LDOC1* was significantly downregulated in OSCC tumors compared with normal oral tissues (*n* = 79; Figure 2c). However, oral tumors displayed an obvious increase in IL-1β mRNA expression compared with noncancerous oral tissues (Figure 2c). An inverse correlation between mRNA expression of *LDOC1* and IL-1β in OSCC and peritumoral oral tissues supports a potential role of *LDOC1* in the production of microbe-induced IL-1β.

### 2.3. CA Infection Promoted the Development of OSCC in the Synthetic Tobacco-Related Carcinogen 4-NQO and the Areca Nut Alkaloid Arecoline-Cotreated Mice

To assess the role of *CA* in oral malignant transformation, we used an OSCC mouse model, established through co-induction of 4-NQO and arecoline (NA), and studied whether the interplay between OSCC-associated carcinogens and *CA* accelerated the development of OSCC [24]. Figure 3a is a schematic representation of the animal study design. Except for the control mice group was not fed with NA (C0 group), all mice were given water containing NA for 8 or 10 weeks (C8 and C10 groups, respectively). Oral lesions were created by brushing the tongue to mimic the trauma generated by betel quid chewing. Starting from week 8, once a week for 8 weeks mice were either painted on the site of tongue lesions with a brush adhered with live *CA* SC5314 (1 × 10^6^ in 20 μL of 0.9% NaCl) (F+) or painted with normal saline as a vehicle control (F−). We collected oral swabs at week 8, 16, and 28 and tested for *CA* infection by analysis using CHROMagar Candida plates as described in Materials and Methods. *CA* infection was determined with the oral swabs obtained at week 28. As shown in Figure 3b, *CA* SC5314 infection was not observed in mice without treatment of NA (C0 group). For the C8F+ and C10F+ groups, the infection rate was correlated with the period for combined treatment of NA. After sacrifice, GMS staining was carried out with the sections of tongue specimen to cross-validate for *CA* infection. The representative GSM staining images of the *CA*-positive samples from C8F+ and C10F+ groups are shown (the right panels of Figure 3b). These data suggest a correlation between NA exposure and *CA* infection. All mice were euthanized at the end of week 28, and the histological slides of tongue tissues were subjected to H&E staining for pathological analysis. Summary statistics based on the incidence of specific pathological features in each group are illustrated in Figure 3c and Table 1. The development of OSCC was observed in all NA-cotreated groups, namely the C8 group (40% and 30% for F- and F+ groups, respectively) and the C10 group (30% and 90% for F− and F+ groups, respectively), but was not observed in the C0 group. These results suggest that the presence of *CA* alone was not sufficient to induce oral carcinogenesis, but oral infection of *CA* acts as a promoting factor of OSCC in mouse with long-term NA exposure. Interestingly, mice from the C0F+ group did not show *CA* colonization but developed either moderate (*n* = 3) or severe (*n* = 2) dysplasia, implying that oral lesions repeatedly exposed to *CA* may be harmful, even if they are not infected (Figure 3b). Representative images of H&E staining for tongue biopsies in all pathological categories are presented in Figure 3d.

### 2.4. LDOC1 Is a Negative Regulator of IL-1β Production Induced by CA SC5314

In the groups of mice with long-term NA exposure (C10 groups), oral infection of *CA* increased the incidence of OSCC, while in the NA-untreated mice groups (C0 groups), neither *CA* infection nor OSCC was observed in the oral lesions repeatedly treated with *CA* (Figure 3b,c and Table 1). These results suggest that exposure to NA may be a prerequisite for *CA* to promote the development of OSCC. Our previous work demonstrated that *LDOC1* can be epigenetically silenced by exposure to cigarette smoke condensates [30]. In addition, an inverse relationship between *LDOC1* expression and IL-1β induced by *CA* SC5314 or *FN* was observed in the cell-based assays (Figure 2a,b). We hypothesized that long-term exposure to NA may cause *LDOC1* silence, thereby increased microbe-induced IL-1β promotes oral carcinogenesis. Because most samples from this animal study had poor RNA quality, it was difficult to analyze mRNA expressions of *LDOC1* and IL-1β in these mouse tissues. To investigate the role of *LDOC1* in IL-1β secretion induced by microbes, we generated cell lines either with *LDOC1* knockdown or with ectopic *LDOC1* expression from CGHNK2 (Appendix A) or TW2.6 cell lines (Figure 4a), respectively. Their vector control cell lines (CGHNK2-shCtrl and TW2.6-GFP), were also established. We examined the effect of *LDOC1* on the production of IL-1β induced by *CA* SC5314. The medium levels of IL-1β were measured using a Cytometric Bead Array (CBA) after cell coculture with live *CA* SC5314 (MOI 0.5) for 8, 16, and 24 h. The results indicated that *CA* SC5314 stimulated IL-1β secretion during the period of coculture in all cells tested (Figure 4b,c). The IL-1β secretion in *LDOC1*-deficient cell lines, CGHNK2-shLDOC1, and TW2.6-GFP significantly increased over time. Notably, the levels of *CA* SC5314-induced IL-1β produced by CGHNK2-shLDOC1 cells were much higher than in CGHNK2-shCtrl cells (Figure 4b). The *CA* SC5314-induced IL-1β produced by TW2.6-LDOC1-GFP cells was significantly reduced as compared with TW2.6-GFP cell lines (Figure 4c). These results indicated that *LDOC1* has an inhibitory effect on IL-1β induced by *CA* in CGHNK2 and TW2.6 cell lines.

### 2.5. LDOC1 Deficiency Induces Inhibitory Phosphorylation of GSK-3β^S9^ through Akt/PI3K Activation, and Reduced GSK-3β Activity Controls the Clonogenicity of CGHNK2-shLDOC1 Cells

The human phospho-kinase array was used to clarify the underlying mechanisms by identifying key molecules involved in the *LDOC1*-mediated suppression of IL-1β production in human oral cells. We determined that glycogen synthase kinase-3β (GSK-3β) and protein kinase B (Akt) accounted for two of the top four kinases with enhanced phosphorylation in CGHNK2-shLDOC1 cells compared with CGHNK2-shCtrl cells (Figure 5a). GSK-3β reportedly plays a role in OSCC [35]. This enzyme has also been demonstrated to regulate the inflammatory response caused by microbial pathogens [36,37]. Activated Akt (phospho-Akt^S473^, pAkt^S473^) is the major kinase responsible for inhibitory phosphorylation of GSK-3β^S9^. Accordingly, we proposed that the Akt/GSK-3β axis may be involved in the *LDOC1*-mediated suppression of IL-1β production and anti-cancer effects in OSCC. The substantial increase in pAkt ^S473^ accompanied by increased phosphorylated GSK-3β (pGSK-3β^S9^, an inactive form of GSK-3β) in CGHNK2-shLDOC1 was validated by Western blotting analysis (Figure 5b). Akt can be activated by phosphatidylinositol 3,4,5-trisphosphate (PIP_3_) produced by phosphoinositide 3-kinase (PI3K). Specific inhibitors of Akt (SH-6) and PI3K (LY294002) were applied to determine whether activated Akt and PI3K were responsible for the increased pGSK-3β^S9^ in *LDOC1*-deficient CGHNK2-shLDOC1 and TW2.6-GFP cell lines. As illustrated in Figure 5c, treatment with SH-6 (5 μM) or LY294002 (10 μM) abolished the phosphorylation of Akt and PI3K, reduced pGSK-3β^S9^, and increased GSK-3β in CGHNK2-shLDOC1 and TW2.6-GFP cell lines. These results indicated that activated Akt and PI3K contributed to the increased pGSK-3β^S9^ and decreased active GSK-3β in *LDOC1*-deficient CGHNK2-shLDOC1 and TW2.6-GFP cell lines. To examine whether GSK-3β is a critical downstream effector of *LDOC1* in OSCC, we examined the effect of ectopic expression of GSK-3β on the acquired clonogenicity of CGHNK2 cells by *LDOC1* knockdown. As illustrated in Figure 5d, *LDOC1* knockdown resulted in acquired anchorage-independent growth (clonogenicity) of CGHNK2-shLDOC1 cells, and ectopic GSK-3β expression (CGHNK2-shLDOC1-GSK-3β) significantly suppressed the clonogenicity acquired after *LDOC1* knockdown, suggesting that the function of GSK-3β is crucial for the anti-cancer activities of *LDOC1*.

### 2.6. The PI3K/Akt/GSK-3β Signaling Pathway Is Involved in the CA-Stimulated IL-1β Production by LDOC1 Deficienct Oral Cells.

To investigate the involvement of the PI3K/Akt/pGSK-3β signaling pathway in the *CA*-induced IL-1β production by *LDOC1*-deficient oral cells, we examined whether SC5314 can activate Akt. As illustrated in Figure 6a,b, coculture with *CA* SC5314 increased pAkt^S473^ in all cells tested, with a peak more rapidly appearing in TW2.6-derived cell lines (20 min) than in CGHNK2-derived cell lines (120 min). In CGHNK2- and TW2.6-derived cell lines, *LDOC1* expression significantly attenuated the induction effect of SC5314 on pAkt^S473^. Furthermore, treatment with inhibitors of Akt (SH-6 (2.5 µM and 5 µM for CGHNK2- and TW2.6-derived cell lines, respectively)) and PI3K (LY294002 (10 µM)) markedly reduced the *CA* SC5314-induced production of IL-1β in *LDOC1*-deficient CGHNK2-shLDOC1 (Figure 6c) and TW2.6-GFP cells (Figure 6d). These results indicated that the enhanced PI3K/Akt signaling caused by *LDOC1* downregulation resulted in the augmented IL-1β production stimulated by microorganisms such as *CA*.

To determine whether inactivated GSK-3β contributes to the dysregulated IL-1β response stimulated by *CA* in *LDOC1*-deficient cells, we increased the abundance of active GSK-3β by inducing the expression of a constitutively active mutant GSK-3β^S9A^ in CGHNK2-shLDOC1 cells. By treatment of CGHNK2-shLDOC1-V cells with *CA* SC5314, pGSK-3β^S9^ expression was observed at only 90 min after coculture with *CA* SC5314 (Figure 7a, left panel). GSK-3β phosphorylation at S9 leads to its inactivation through proteasomal degradation [38,39], which may explain the absence of pGSK-3β^S9^ in the CGHNK2-shLDOC1-V cells at most time points in the experiment. In the CGHNK2-shLDOC1-GSK-3β^S9A^ cells stimulated with *CA* SC5314, the protein expression of pGSK-3β^S9^ decreased at 60 min and then increased significantly at 90 min, peaking at 150 min (Figure 7a, center panel). Ninety minutes after *CA* SC5314 stimulation of CGHNK2-shLDOC1-V cells, the expression of active GSK-3β decreased gradually, and only one-tenth of the original amount remained at 150 min (Figure 7a, right panel). In the CGHNK2-shLDOC1-GSK-3β^S9A^ cells stimulated with *CA* SC5314, the total protein expression of GSK-3β and GSK-3β^S9A^, which accounted for the enzyme activity, remained stable without apparent reduction. These results indicated that expressing GSK-3β^S9A^ can compensate for the reduced active form of GSK-3β in response to *CA* SC5314 treatment (Figure 7a). Figure 7b illustrated that *CA* SC5314 (MOI 0.5) can also induce CGHNK2-shLDOC1-GSK3β^S9A^ cells to secrete IL-1β after coculture for 24 h. As compared to those secreted by the corresponding control cell line CGHNK2-shLDOC1-V, the levels of *CA* SC5314-induced IL-1β produced by CGHNK2-shLDOC1-GSK3β^S9A^ significantly reduced. Altogether, these results demonstrated that the PI3K/Akt/GSK-3β axis is involved in the augmented *CA*-induced IL-1β production caused by *LDOC1* deficiency in oral cells. The immune actions affected by GSK-3β are largely attributable to the numerous crucial immune-related transcription factors (TFs) that it regulates, such as NF-kB, CREB, AP-1, STAT1-3, and β-catenin [37,40]. Results from TF DNA-binding profiling assay performed with CGHNK2-shLDOC1 and CGHNK2-shCtrl cells (Figure 7c) demonstrated that up to 96% (23/24) of the total 24 *LDOC1*-mediated TFs identified (5 and 19 had decreased and increased DNA-binding ability by *LDOC* knockdown, respectively) were relevant to immune development and inflammatory responses. These results support that GSK-3β is a crucial acting point for the immunomodulating function of *LDOC1* in human oral cells.

## 3. Discussion

IL-1β has been recognized as a salivary biomarker for OSCC and can promote oral malignant transformation [22,24]. In general, the pathogenesis of OSCC differs between regions and ethnicities. Some studies indicated that *CA* and *FN* are promoting factors in the development of OSCC [7,8,17]. However, published studies on the association between *CA* and OSCC require further investigation of cases from Taiwan. Two specific reasons motivated our experiment, as illustrated in Figure 1. First, *LDOC1* is a tumor suppressor gene that was identified in OSCC specimens in Taiwan, and second, we primarily used OSCC cell lines generated from Taiwan OSCC tissues to conduct the subsequent cell-based functional analysis of *LDOC1* in OSCC. Specifically, we decided to initially test the proportion of *CA*-positive oral specimens in OSCC cases from Taiwan (Figure 1a,b), and, for consistency, we also verified that *CA* SC5314 invaded TW2.6 and CGHNK2 cell lines, which were two cell lines established from the tissues of Taiwanese patients with OSCC (Figure 1c). Therefore, the results in Figure 1 are novel in that they elucidate the roles of *LDOC1* and *CA* in OSCC in Taiwanese patients. In this study, we demonstrated that *CA* and *FN* are potent inducers of IL-1β in human normal and cancerous oral cells, especially for cells deficient in *LDOC1* expression (Figure 2a,b). We analyzed IL-1β in the conditioned medium of tested cells. We did so in a direct contact coculture system where the human oral cells were cultured in a growth medium containing either L or HK *CA* preparations. Therefore, even if HK *CA* cannot invade oral cells, HK *CA* still has direct contact with the cells. Previous studies have shown that fungal β-glucan (glucose polymers found in the cell walls of fungi) is key to immune recognition in macrophages; it stimulates the production of inflammatory cytokines, mainly through Dectin-1, which is a β-glucan-specific membrane receptor. Recent reports have shown that surface mannoproteins can mask the underlying β-glucan from the receptors. *CA* cells expose their masked β-glucan after being heat killed. In doing so, heat-killing treatment facilitates the interaction between Dectin-1 and the exposed β-glucan, and, in turn, internalization of β-glucans mediates the production of proinflammatory cytokines such as IL-1β and IL-8. Dectin-1 expresses on the cell surface of phagocytes as well as in normal and cancerous oral cells (data not shown). Our results were consistent in indicating that the HK *CA*-induced IL-1β production was nearly abolished by pretreatment with Dectin-1-neutralizing antibodies (data not shown). *LDOC1* as a negative regulator of *CA*-induced IL-1β production (Figure 4b,c) and this inhibitory effect involved PI3K/Akt/GSK-3β signaling (Figure 6 and Figure 7). This stimulation effect can be nearly abolished or markedly reduced by *LDOC1* expression. Inhibitors of PI3K and Akt (Figure 6c,d), and increased GSK-3β activity (Figure 7b), attenuated the augmented production of *CA*-stimulated IL-1β in *LDOC1*-deficient oral cells, whereas this suppressing effect was lower in *LDOC1*-expressing cells, suggesting that the PI3K/Akt/GSK-3β signaling pathway is involved in *LDOC1*-mediated microbe-induced IL-1β production. We proposed a model based on our data for the role of *LDOC1* in OSCC (Figure 8). *LDOC1* being silenced or downregulated by promoter hypermethylation in smokers causes the activation of PI3K and Akt, which reduces the activities of GSK-3β and thus results in IL-1β response aberrations. In this setting, human oral cells secrete high levels of IL-1β to promote malignant transformation of oral lesions in the presence of microbes, such as *CA* and *FN*.

Several lines of evidence have indicated that numerous tumor suppressor genes, such as p53, PTEN, RB1, and ARF, play important roles in the immune system [41]. Therefore, tumor suppressor genes may act not only as guardians of genomic integrity but also as critical factors in the homeostatic regulation of immune response. Findings from this study revealed that *LDOC1* is crucial for the IL-1β response to microbial exposure. Cigarette smoke modulates both the innate and adaptive immune systems. In addition to activating macrophage and dendritic cell activity, tobacco smoke exposure increases the production of numerous proinflammatory cytokines, such as IL-1, IL-6, IL-8, TNF-α, and GM-CSF, and reduces the levels of anti-inflammatory cytokines, such as IL-10 [42]. *LDOC1* being epigenetically silenced by tobacco exposure [30] may explain the effect of smoking on immunity and inflammation [42]. Altered immunity can cause commensal microorganisms to become pathogenic and induce high levels of oncogenic IL-1β for oral cancer. This assumption is indirectly supported by observations from epidemiologic studies that have indicated that poor oral hygiene, which may cause proliferation of oral microorganisms, is an independent risk factor for OSCC, particularly among cigarette smokers and alcohol users [17,26].

The mechanisms that control GSK-3β activity are not yet fully understood. However, GSK-3β activity is thought to be controlled, in large part, by phosphorylation. GSK-3β is constitutively active under basal cellular conditions. Inhibitory phosphorylation of GSK-3β^S9^ can be triggered by cellular stimuli, such as those received through growth factor receptors, Toll-like receptors, T cell receptors, and interleukin receptors. Our results revealed that the activated Akt and PI3K are two key molecules controlling the activities of GSK-3β (Figure 5c), and that PI3K/Akt/GSK-3β signaling significantly affects IL-1β production in *LDOC1*-deficient cells (Figure 6c,d and Figure 7b), indicating that *LDOC1* at least partly regulates immune actions by modulating the activities of GSK-3β via Akt and PI3K. The activation of Akt and PI3K are tightly regulated by ubiquitination. Ubiquitination is a post-translational modification by the covalent attachment of the small polypeptide ubiquitin that alters the properties of its target proteins in a variety of manners and serves as a valuable signal in the host immune response to pathogenic infection. In eukaryotes, the conjugation of ubiquitin to target proteins involves the consecutive actions of three enzymes, the ubiquitin-activating enzyme E1, the ubiquitin-conjugating enzyme E2, and the ubiquitin ligase E3 [43]. Ubiquitination is a reversible modification. Ubiquitin can be removed from its targets by deubiquitinating enzymes (DUBs). Lin’s research group discovered that Akt also undergoes K63-linked ubiquitination by the E3 ligase TNF receptor-associated factor (TRAF) 6 upon cellular stimulation with growth factors or cytokines inducing Akt activation [44]. This K63-linked ubiquitination of Akt does not alter the protein stability of Akt, but it is crucial for membrane recruitment and subsequent activation of Akt. Furthermore, pAkt associates with the E3 ligases TTC3 and MULAN to facilitate ubiquitination and degradation within the nucleus [45,46]. These findings reflect the profound influence of ubiquitination on Akt activity. PI3K also undergoes ubiquitination [47,48]. We identified two E3 ligases as potential LDOC1-interacting partners by searching the STRING database (http://string-db.org). Therefore, *LDOC1* may negatively regulate the activation of Akt and PIK3 through ubiquitination. Our data support the assumption that *LDOC1*-deficient and *LDOC1*-expressing cells exhibited different kinetic patterns of Akt phosphorylation in oral premalignant and OSCC cell lines in response to SC5314 stimulation (Figure 6a,b).

Activation of NF-kB and the assembly of inflammasomes are two essential signals for the production and secretion of IL-1β [49]. *LDOC1* may also control the production of IL-1β through the ubiquitination of NF-kB and inflammasome. Inflammasome is composed of three major components, pattern recognition receptors (PRRs), an adaptor protein (PYCARD or ASC), and procaspase-1 or procaspase-5 [49]. When PRRs sense pathogen-associated or danger-associated molecular pattern signals, the PRRs interact with PYCARD, which recruits and activates procaspase-1 or procaspase-5. NLRP3 inflammasome plays an essential role in host defense against bacterial infection, and its regulation has been extensively studied. Increasing evidence indicates a central role for ubiquitination in NLRP3 inflammasome regulation [50]. The Alnemri group demonstrated that the deubiquitylation of NLRP3 by DUB BRCC3 was required for NLRP3 inflammasome activation [51]. Furthermore, David Brough et al. identified that DUB USP7 and USP47 regulate NLRP3 inflammasome activation by promoting ASC oligomerization and speck formation [52]. Regulation of the NF-κB activation pathway by the ubiquitin proteolytic system is well documented [53]. PRRs can activate the canonical NF-κB pathway. The primary mechanism for canonical NF-kB activation is the inducible degradation of IκBα, triggered by site-specific phosphorylation by the IκB kinase (IKK) complex. The IKK comprises two catalytic subunits, IKKα and IKKβ, and a regulatory subunit, NF-κB essential modulator (NEMO). IKK is activated by stimulation with various agents, such as inflammatory cytokines, mitogens, growth factors, and microbial components. Upon activation, IKK phosphorylates IκBα, which is then conjugated with K48-linked polyubiquitin chains by the E3 ligase SCF^β-TRCP^ and degraded by the proteasome, resulting in the release of NF-κB from IκBs and translocation to the nucleus to induce the transcription of target genes. These findings emphasize the central role of ubiquitination in NF-kB and inflammation regulation. Inhibition of PI3K/Akt/pGSK-3β signaling reduced the production of microbe-induced IL-1β but failed to fully compensate for the augmented secretion of IL-1β caused by *LDOC1* loss (Figure 6c,d and Figure 7b). These results suggested that, in addition to the regulation of PI3K/Akt/pGSK3β signaling, *LDOC1* may suppress the biosynthesis of IL-1β by modulating the ubiquitination efficiency for components of NF-kB and inflammasome. Future studies should investigate the effects of *LDOC1* on the activities of these critical molecular modules through ubiquitination.

## 4. Materials and Methods

### 4.1. Clinical Samples

Paired tumor specimens and their surrounding non-cancerous tissues were obtained from 80 previously untreated patients who were diagnosed during (2013–2016) at the Chi Mei Medical Center in Tainan, Taiwan. Fresh-frozen tissues were preserved at −80 °C at the biobank (CMBEC-106-009) of the hospital and used to prepare tissue sections for GMS staining and *CA*-specific real-time PCR assay. This study was approved by the Institutional Review Board (IRB) of Chi Mei Medical Center and National Health Research Institutes, Taiwan with IRB approval numbers 1060-006 (15 December 2017–14 December 2018) and EC1050109-E (1 August 2016–31 July 2017), respectively. Informed consent was obtained from each patient.

### 4.2. Antibodies and Reagents

For Western blotting analysis, antibodies against AKT (GTX121937), and GAPDH (GTX100118) were purchased from GeneTex (Hsinchu, Taiwan); anti-pAKT^S473^ (CST4060), anti-GSK-3β (CST9315), and anti-pGSK-3β^S9^ (CST9323) were purchased from Cell Signaling (Danvers, MA, USA). The PI3K inhibitor LY294002 and Akt inhibitor SH-6 were obtained from Enzo Life Sciences (Farmingdale, NY, USA). *CA* strains SC5314 and SC5314-GFP were gifts from Dr. Lo Hsiu-Jung (National Institute of Infectious Diseases and Vaccinology, National Health Research Institutes, Taiwan). A plasmid containing cDNA encoding a GSK-3β mutant GSK-3β^S9A^ was a gift from Scott Friedman lab (Addgene plasmid # 49492; http://n2t.net/addgene:49492; RRID:Addgene_49492). The cDNAs was subcloned into pLentiviral vector to generate lentivirus carrying GSK-3β^S9A^ cDNA. Lentiviral pGIPZ carrying either shRNA vectors targeting human *LDOC1* (shLDOC1) or GFP-tagged *LDOC1* ORF were purchased from GE Dharmacon (Lafayette, CO, USA).

### 4.3. Cell Lines and Transduction

The immortalized normal oral keratinocyte cell line CGHNK2 was a gift from Dr. Cheng, Ann-Joy (Chang Gung University, Taoyuan, Taiwan) and was grown in keratinocyte serum-free medium (KSFM) with supplements (EGF, Human Recombinant/Bovine Pituitary Extract from Life Technologies, Inc., Gibco BRL, Rockville, MD, USA, respectively) as previously described [26,47]. TW2.6 has been established from the surgically resected specimen of an untreated primary squamous cell carcinoma of the buccal mucosa from a patient who was an betel quid chewer and tobacco smoker, and was cultured as previously described [27,29]. In vitro transduction was carried out at one multiplicity of infection (MOI) per cell. To generate stable cell lines, the lentivirus-infected cell pools were selected with either 5 µg/mL puromycin (Sigma-Aldrich, St. Louis, MO, USA) or 20 μg/mL blasticidin S (Invitrogen, Waltham, MA, USA).

### 4.4. Quantitative Real-Time PCR (qPCR)

Total RNA was isolated using RNeasy Mini Kits (Qiagen, Hilden, Germany) according to the manufacturer’s instructions. The synthesis of cDNA from total RNA was performed with M-MLV reverse transcriptase (Promega, Madison, WI, USA). The resulting cDNAs were subjected to qPCR using the following primers: 5′-ATGACGACGAAGACGACGA-3′ (F) and 5′-GAGGGTCGAGGGCCTAATAA-3′ (R) for *LDOC1*. QPCR was performed with FastStart Universal Probe Master Kit (Roche Applied Science, Madison, WI, USA). The gene expression level was normalized using GAPDH mRNA.

### 4.5. Detection of Fungal Organisms Using Grocott’s Methenamine Silver (GMS) Staining

For fungal microorganism examination, deparaffinized and hydrated oral tissue sections were stained with Grocott methenamine silver (GMS) stain kit (ScyTek Laboratories, Inc., West Logan, UT, USA) following manufacturer’s instructions. Briefly, GMS is a chromic acid which oxidizes fungal cell wall mucopolysaccharide components to form aldehyde groups. The aldehyde groups present in the fungal cell wall then react with the silver nitrate, reducing it to metallic silver, which appears black. The stained samples were observed under a light microscope (Olympus-Japan BX 41, magnification × 400, Tokyo, Japan). The fungi were turned black, while everything else is stained light green.

### 4.6. Detection of CA DNA Using Real-Time PCR Assay

DNA extracted from tissue sections using QIAamp DNA FFPE tissue kit (Qiagen 56404) following the manufacturer’s instructions. The PCR amplicon specific for internal transcribed space of *CA* rDNA is 108 base pairs in size using the primer sequences and probe as described previously [54]. Real-time PCR was conducted using PowerUp™ SYBR Green Master Mix using an ABI 7500 fast Real-time PCR System (Applied Biosystems, Foster City, CA, USA). Each reaction was run in duplicate and contained 100 ng gDNA template along with 50 nM primers in a final reaction volume of 20 µL. The cycling parameters began with 95 °C for 2 min, followed by 40 cycles of 95 °C for 3 s and 60 °C for 30 s. Samples were also run a 3% agarose gel to confirm specificity.

### 4.7. Mouse OSCC Model with CA-Infected Oral Lesions

The animal experimental protocol was approved by Institutional Animal Care and Use Committee. C57BL/6JNarl mice (8 weeks of age) were purchased from the National Laboratory Animal Center (Taipei, Taiwan). The mouse OSCC was induced with combination of 4-NQO and arecoline (NA) as described previously [24]. Briefly, mice were given water containing NA for 8 (C8) or 10 (C10) weeks. Mice without NA treatment were used as control group (C0). Oral lesions were created by brushing the tongue to mimic the trauma generated by betel quid chewing. Starting from week 8, once a week for 8 weeks mice were either painted with live *CA* SC5314 at the site of lesions on their tongue (F+) or painted with normal saline as a vehicle control (F−). All mice were euthanized at the end of week 28 (day 196) and the tongue sections were carefully collected immediately after euthanasia. The histological slides of tongue tissues were processed for H&E staining and GSM staining for histopathological and fugus examination, respectively.

### 4.8. Determination of CA Infection in Mouse Oral Lesions

*CA* infection was examined from oral swabs collected at week 8, 16, and 28. Oral infection of *CA* was assessed using CHROMagar™ Candida plate (CHROMagar™, Paris, France). Briefly, samples were collected by gently rubbing a sterile cotton swab over the mouse oral cavity and were immediately transferred for microbiological analysis. The collected oral swabs were soaked in PBS and shaken for one minute, and then the PBS was streaked onto CHROMagar plates and incubated at 37 °C for 24 h. *CA* infection was determined with the oral swabs obtained at week 28. Positive for *CA* infection was defined as the appearance of creamy and convex pasty colonies with a moldy odor. No colonies observed even after 24 h of incubation was recognized as negative. After sacrifice, the section of mouse tongue specimen was subjected to GMS staining to cross-validate for *CA* infection. The infection rate is defined as the rate of *CA*-positive mice to the total number of mice in an experimental group.

### 4.9. Cytometric Bead Array (CBA) Analysis

The production of IL-1β in the culture supernatant was determined by a Human Inflammation CBA Kit (BD Biosciences, CA) following the manufacturer’s instructions. Conditioned medium was prepared from 10^5^ cells seeded onto a 6 cm dish and cultured in 2 mL of serum-free medium in the presence or absence of preparations of *CA* SC5314 or *FN*. The BD FACSCalibur flow cytometer was used to acquire data by setting the red diode laser for exciting the CBA particles and detection of particle emission in the FL4 channel.

### 4.10. Gene Expression Microarray Analysis

Extraction of total RNA, RNA quality evaluation, and cDNA microarray experiments were performed according to Affymetrix standard protocols by the microarray core laboratory at the National Health Research Institutes. The gene chips (Clariom S Assay HT06, Affymetrix Inc, Santa Clara, CA, USA) were scanned with an Affymetrix Gene ChIP Scanner 3000 7G, and the CEL files generated were analyzed using Affymetrix Expression Console Software (version 1.4, Affymetrix, Santa Clara, CA, USA), which normalizes array signals using Signal Space Transformation (SST) and a robust multiarray averaging (RMA) algorithm. Normalized data were analyzed using Transcriptome Analysis Console (TAC) 3.0 software (Affymetrix, Santa Clara, CA, USA). A paired *t*-test was applied to identify differentially expressed transcript genes between sample pairs and probes, with *P*-values less than 0.05 and fold-change ≥2 declared significant. All gene level files were imported into QIAGEN’s Ingenuity Pathway Analysis (IPA) software for pathway and molecular function analysis. Microarray expression data are available at the U.S. National Center for Biotechnology Information Gene Expression Omnibus (GEO) database under accession number GSE149108.

### 4.11. Human Phospho-Kinase Array Analysis

Total protein concentrations in cell lysates were measured using a NanoDrop ND-1000 spectrophotometer (Wilmington, DE, USA). Four hundred micrograms of cell lysates were hybridized with the Proteome Profiler Human Phospho-Kinase array (ARY003B, R&D systems, Minneapolis, MN, USA) according to the manufacturer’s instructions. Chemi Reagent Mix was applied for visualizing the hybridized spots, then spot densities were quantified using ImageJ v1.52.

### 4.12. Transcription Factor (TF) Profiling Array

Experiments were performed as previously described [31]. Briefly, cellular nuclear proteins were harvested using a Nuclear Extraction Kit (Chemicon, Rolling Meadows, IL, USA). Experiments were carried out with The TF Activation Profiling Plate Array system (Signosis, Santa Clara, CA, USA) according to the manufacturer’s instructions. The DNA binding activities of TFIID were used for normalization.

### 4.13. Soft Agar Assay

Experiments were performed and quantified as previously described [24].

### 4.14. Statistics

Student tests and ANOVA tests were analyzed using Microsoft Excel (Microsoft, Redmond, WA, USA) and statistical software NCSS (NCSS, Kaysvill, UT, USA), respectively.

## 5. Conclusions

This study highlighted that *LDOC1* acts as a novel IL-1β modulator to control inflammatory responses to oral microbes, such as *CA* and *FN*. Furthermore, we identified molecules, such as PI3K, Akt, and GSK-3β, that acted as key players in the *LDOC1*-mediated IL-1β response. Chronic inflammation that is characterized by a sustained level of IL-1β may create a protumor microenvironment for OSCC. Accordingly, to prevent OSCC, we recommend smoking cessation to avoid *LDOC1* silencing, which can cause immune abnormalities, and good dental hygiene so as to not allow proliferation of microbes, such as *CA* or *FN*, in oral cavities.

## Figures and Tables

**Figure 1 cancers-12-03148-f001:**
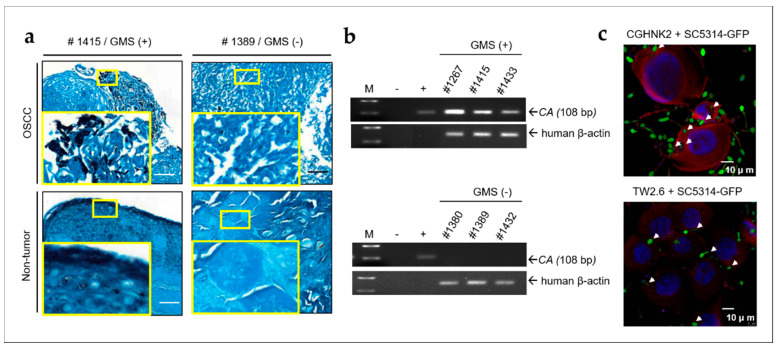
*CA* (*Candida albicans*) is present in oral specimens from patients with OSCC (oral squamous cell carcinoma) and invaded CGHNK2 and TW2.6 cell lines. (**a**) GMS (Gomori methenamine silver) stain showing the presence of fungi in OSCC tumor and adjacent non-tumor tissues. Fungus was black stained. The number in each picture indicates the sample code. GMS-negative samples were shown as controls. Small and large yellow frames indicate the original and magnified areas, respectively. Bar, 100 μm. (**b**) Gel views of PCR products obtained with DNA extracted from OSCC biopsies. M, molecular weight marker (100-bp ladder); −, negative control; +, PCR carried out with DNA isolated from *CA* SC5314 as positive control. The size of the *CA* PCR product is indicated by an arrow (108 bp). Human β-actin PCR product used as internal control for each OSCC samples. (**c**) *CA* invade cultured human untransformed CGHNK2 and OSCC TW2.6 cell lines. Immunofluorescence confocal micrograph of CGHNK2 and TW2.6 cells after coculture with *CA* strain SC5314 expressing GFP (SC5314-GFP, MOI of 100) for 1 h. Invading intracellular *CA* (green) were confirmed by the single optical section through the host cells and indicated by white arrows. Fixed CGHNK2 and TW2.6 cells were stained with phalloidin-TRITC (red) to show filamentous actin. The nuclear counter stain is DAPI (4′,6-diamidino-2-phenylindole, blue). Bar, 10 μm.

**Figure 2 cancers-12-03148-f002:**
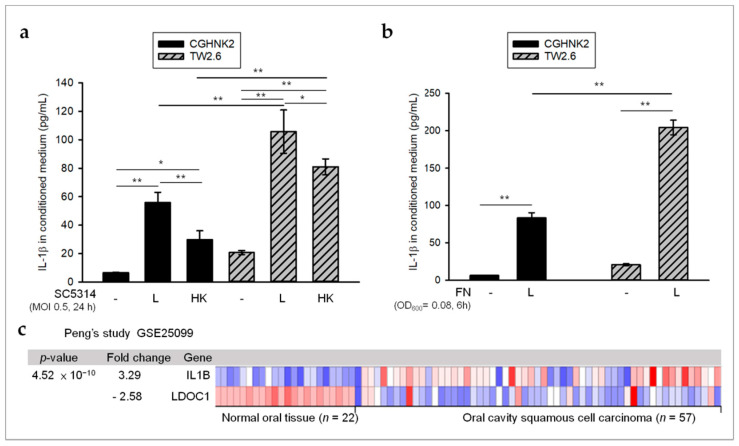
Inverse relationship between *LDOC1* and *CA* SC5314- or *FN*-induced IL-1β in OSCC. (**a**,**b**) *LDOC1*-deficient TW2.6 cells secrete more microbe-induced IL-1β as compared with *LDOC1*-expressing CGHNK2 cells. Cytometric bead array (CBA) analysis of IL-1β in conditioned medium of CGHNK2 and TW2.6 cell lines without microbes or with either live (L) or heat-killed (HK) *CA* SC5314 (MOI 0.5) (**a**) and L *FN* (OD_600_ = 0.08) (**b**) for 24 and 6 h, respectively. Results are presented as concentrations (pg/mL). Data are representative of at least three independent experiments (mean + S.D.), analyzed using the ANOVA test. * *p* < 0.05 and ** *p* < 0.01. (**c**) The mRNA expression profiles of LDOC1 and IL-1B in OSCC tumors (*n* = 57) and normal oral tissues (*n* = 22) were obtained from publicly available microarray data sets (GSE25099) [34] in Oncomine (https://www.oncomine.com).

**Figure 3 cancers-12-03148-f003:**
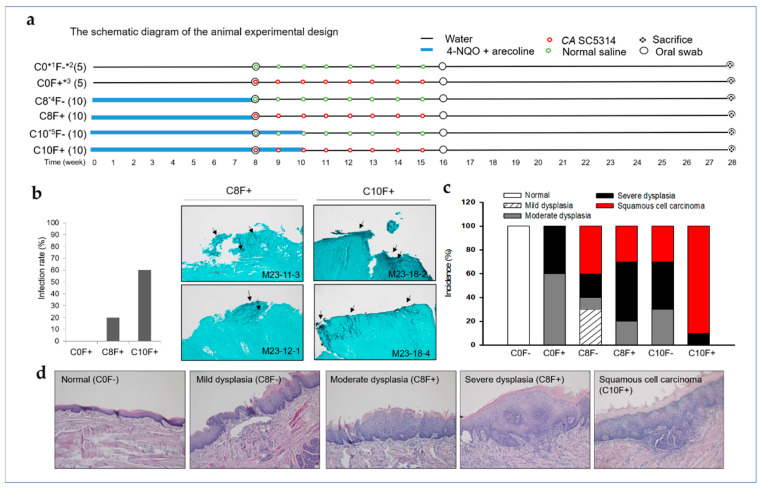
Oral infection of *CA* promotes the development of OSCC in mouse cotreated with 4-NQO and arecoline (NA). (**a**) The schematic diagram of the animal experimental design. (**b**) Histogram of *CA* SC5314 oral infection rate in mice groups. Oral swabs were collected at week 8, 16, and 28 and sent for microbial culture using CHROMagar Candida plates. Oral infection of *CA* SC5314 was determined with the oral swabs obtained at week 28. Representative GSM staining images of the *CA*-positive samples from C8F+ and C10F+ groups are shown. Arrows indicate the presence of fungi (black stained) in the samples. Magnification, ×100. (**c**) The bar chart plots percentage of pathological categories in tongue biopsies collected from mice in each group. (**d**) Representative histopathological sections of tongue tissues with different stages of malignant transformation. Magnification, ×100. *^1^ The control mice group without fed with NA; *^2^ Without treatment of *CA* SC5314; *^3^ With treatment of *CA* SC5314; *^4^ Mice group fed with water containing NA for 8 weeks; *^5^ Mice group fed with water containing NA for 10 weeks.

**Figure 4 cancers-12-03148-f004:**
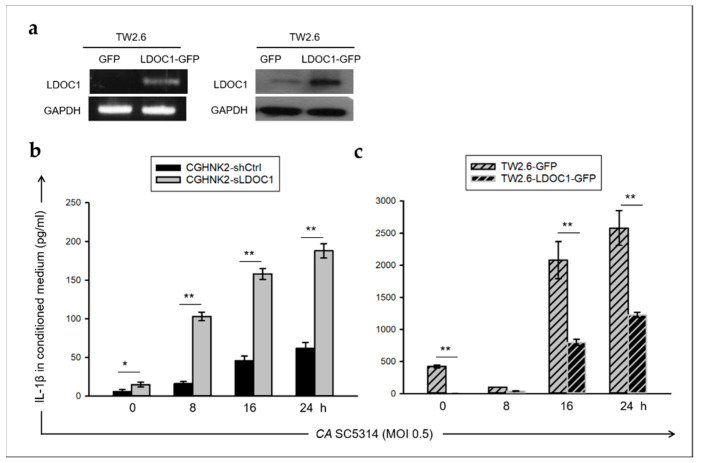
*LDOC1* suppresses IL-1β production upon *CA* stimulation in CGHNK2 and TW2.6 cell lines. (**a**) Expression of LDOC1 mRNA (left panels) and protein (right panels) were ectopically expressed in TW2.6 cell line. The TW2.6 cell line was transduced lentivirus carrying either control vector (TW2.6-GFP) or ORF encoding *LDOC1* (TW2.6-LDOC1-GFP). (**b**,**c**) Time course of IL-1β production. CBA (Cytometric Bead Array) was used to detect secreted IL-1β in the conditioned medium of CGHNK2- (**b**) and TW2.6- (**c**) derived cell lines after stimulation with *CA* SC5314 (MOI 0.5) for indicated time. Three independent experiments were performed, each with three duplicates. The representative results from one experiment are shown. Data are presented as mean ± SD (*n* = 3), analyzed using the ANOVA test. * *p* < 0.05 and ** *p* < 0.01.

**Figure 5 cancers-12-03148-f005:**
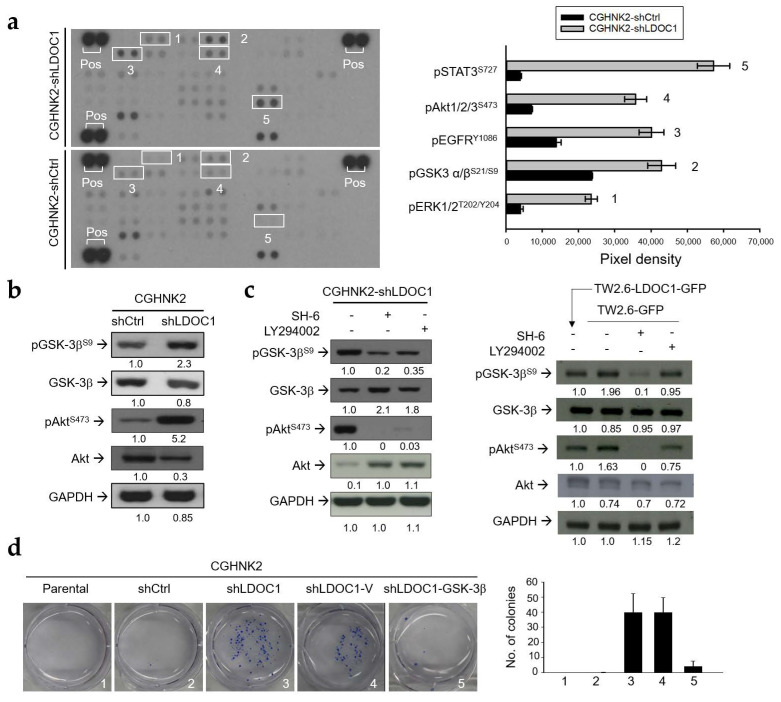
*LDOC1* downregulation leads to inhibitory phosphorylation of GSK-3β^S9^ by pAkt^S473^ and GSK-3β regulates the clonogenicity of the CGHNK2-shLDOC1 cell line. (**a**) Results of the human phosphokinase array analysis using CGHNK2-shLDOC1 and CGHNK2-shCtrl cells. The bar charts represent the relative protein phosphorylation levels for the top four kinases and the transcription factor STAT3 exhibiting different phosphorylation states in CGHNK2-shLDOC1 and CGHNK2-shCtrl cells (the array contained some non-kinase phosphoprotein spots). Experiments were performed according to the instructions provided by the manufacturer, as described in Materials and Methods. (**b**) Western blotting analysis of pGSK-3β^S9^, GSK-3β, pAkt^S473^, and Akt in CGHNK2-shLDOC1 and CGHNK2-shCtrl cells. GAPDH was used as a loading control for each protein sample. (**c**) The serine kinase activity of Akt is required for the phosphorylation of GSK-3β^S9^ in CGHNK2-shLDOC1 and TW2.6-GFP cells. Cellular protein lysate was harvested after a 6 h treatment with inhibitors of Akt (SH-6, 5 μM) or PI3K (LY294002, 10 μM). Western blotting analyses of pGSK-3β^S9^, GSK-3β, pAkt^S473^, Akt, and GAPDH were then performed. Cells treated with PBS (0.1%) were used as controls. The pixel intensity ratio for each band was shown. There are technical duplicates for the human phosphokinase array analysis; Western blotting analysis under each condition was performed at least twice. (**d**) Ectopic expression of GSK-3β inhibits the acquired clonogenicity of CGHNK2-shLDOC1 cells. Cells were subjected to a colony-forming assay. The bar chart represents the mean values of triplicate tests (mean ± SD). Representative images are presented.

**Figure 6 cancers-12-03148-f006:**
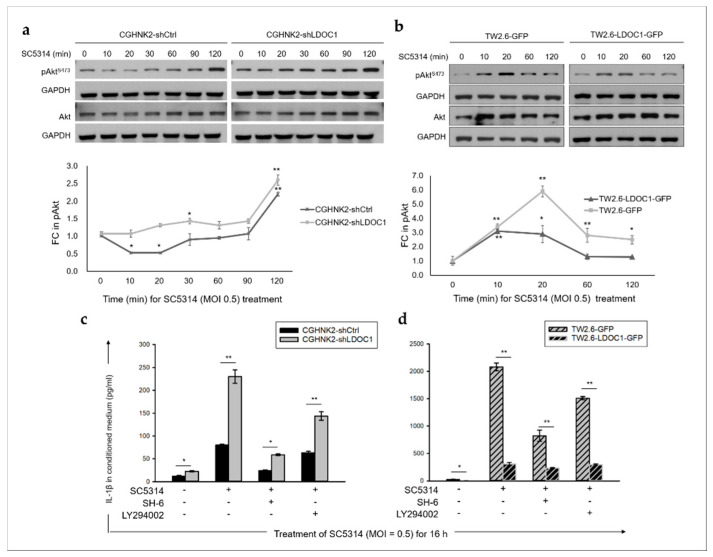
**The** PI3K/Akt signaling pathway involved in *CA*-induced IL-1β production in *LDOC1*-deficient oral cells. (**a**,**b**) Effect of *CA* SC5314 on the phosphorylation of Akt^S473^ in *LDOC1*-deficient or *LDOC1*-expressing oral cell lines. Cells cultured in a normal growth medium were treated with *CA* SC5314 (MOI = 0.5) for the times indicated. Equal amounts of whole-cell lysates were subjected to Western blotting analysis with antibodies specific for pAKT^S473^, Akt, and GAPDH. A representative blot is presented in the upper panels. As presented in the bottom panels, densitometric analyses were performed to quantify the fold change (FC) in the intensity of the pAKT^S473^ blots with untreated controls (time 0) set as 1. Data are expressed as mean ± SD (*n* = 2, biological duplicates). * *p* < 0.05, ** *p* < 0.01 vs. basal activation. (**c**,**d**) Levels of IL-1β induced by *CA* SC5314 were suppressed by inhibitors of Akt and PI3K (SH-6 and LY294002, respectively) in *LDOC1*-deficient CGHNK2-shLDOC1 (**c**) and TW2.6-GFP (**d**) cell lines. Concentrations of IL-1β in the conditioned medium of cells were measured by CBA after coculture with live *CA* SC5314 for 16 h, with or without pretreatment of SH-6 (2.5 µM and 5 µM for CGHNK2- and TW2.6-derived cell lines, respectively) or LY294002 (10 µM) for 6 h. Data are presented as mean ± SD (*n* = 3), analyzed using the ANOVA test. * *p* < 0.05 and ** *p* < 0.01.

**Figure 7 cancers-12-03148-f007:**
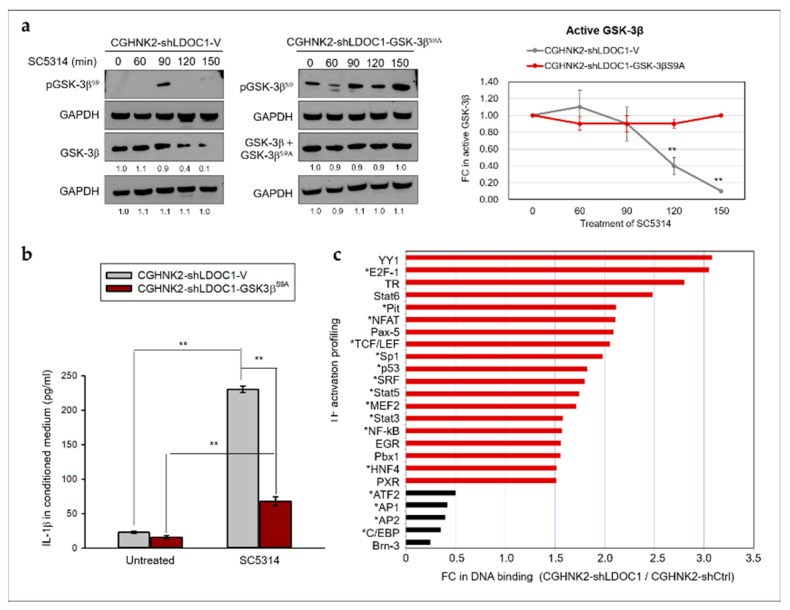
GSK-3β participates in the enhanced IL-1β production in *LDOC1*-deficient CGHNK2-shLDOC1 cells after coculture with *CA* SC5314. (**a**) Western blotting analyses of the quantities of active GSK-3β, GSK-3β^S9A^, and inactive pGSK-3β^S9^ proteins in CGHNK2-shLDOC1-V (left panel) and CGHNK2-shLDOC1-GSK-3β^S9A^ (center panel) cell lines. Densitometric analyses were performed to quantify the FC in the intensity of the active non-phosphorylated GSK-3β blots with untreated controls (time 0) set as 1 (right panel). Data are expressed as mean ± SD (*n* = 2, biological duplicates). ** *p* < 0.01 versus basal activation. (**b**) Concentrations of IL-1β in the conditioned medium were measured as described in the caption to Figure 4; three independent experiments were performed for IL-1β, each with three duplicates. Data are presented as mean ± SD (*n* = 3), analyzed using the ANOVA test. ** *p* < 0.01; (**c**) *LDOC1* modulates immune-related TFs. Nuclear proteins isolated from CGHNK2-shCtrl and CGHNK2-shLDOC1 cells were subjected to a TF DNA-binding profiling assay. The fold change (FC) of intensities derived from TFII in CGHNK2-shLDOC1 versus CGHNK2-shCtrl was used for normalization. Experiments were performed according to the instructions provided by the manufacturer, as described in Materials and Methods. The bar charts represent the average FC values of two independent experiments.

**Figure 8 cancers-12-03148-f008:**
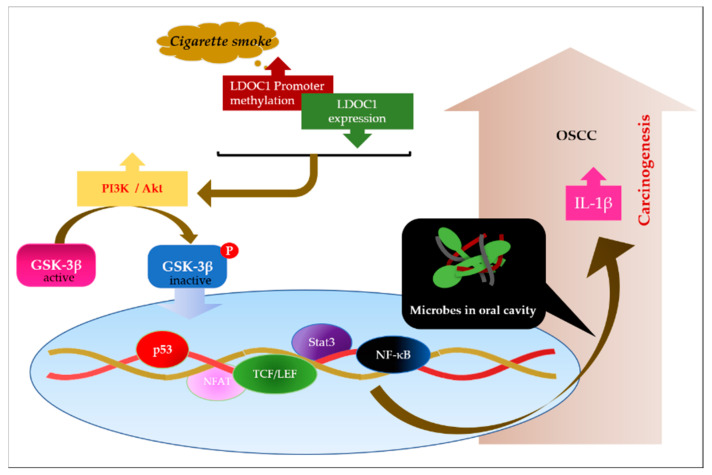
Model outlining the mechanism through which *LDOC1* functions as an immune-modulator to attenuate the IL-1β production increased upon microbial infection of human oral cells.

**Table 1 cancers-12-03148-t001:** Histopathological changes in the tongue tissues of mice in each group.

Group (*n*)	No. (%) of Biopsies						
	Normal	Squamous Hyperplasia	Mild Dysplasia	Moderate Dysplasia	Severe Dysplasia	Squamous Cell Carcinoma	Survival *
C0F− (5)	5 (100%)	0	0	0	0	0	5 (100%)
C0F+ (5)	0	0	0	3 (60%)	2 (40%)	0	5 (100%)
C8F− (10)	0	0	3 (30%)	1 (10%)	2 (20%)	4 (40%)	8 (80%)
C8F+ (10)	0	0	0	2 (20%)	5 (50%)	3 (30%)	8 (80%)
C10F− (10)	0	0	0	3 (30%)	4 (40%)	3 (30%)	9 (90%)
C10F+ (10)	0	0	0	0	1 (10%)	9 (90%)	10 (100%)

“*”: survival before sacrifice.

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
