# Peer review of "LDOC1 Suppresses Microbe-Induced Production of IL-1β in Human Normal and Cancerous Oral Cells through the PI3K/Akt/GSK-3β Axis"

_cancers, 2020, doi:10.3390/cancers12113148_

Round 1
Reviewer 1 Report
The authors previously reported that LDOC1 (leucine-zipper downregulated in cancer 1), an X-linked tumor suppressor, is frequently silenced by promoter hypermethylation in OSCC (oral squamous cell carcinoma) patients who habitually smoke cigarettes. In this work, they explored the major function of LDOC1 in human oral cells and reports that LDOC1 suppresses Candida albicans- and Fusobacterium nucleatum-induced production of oncogenic IL-1β by normal and cancerous oral cells through the PI3K/Akt/GSK-3β axis. The manuscript contains useful information on malignant transformation of oral cells. I have no serious comments on this manuscript. During the review, however, I noticed several minor points to be considered by the authors. These are listed below (suggestions are indicated by "-->").
p. 1, lines 2-4:
LDOC1 Suppresses the Production of Microbes-induced IL-1β in Human Normal and Cancerous Oral Cells through PI3K/Akt/GSK-3β Axis
--> LDOC1 Suppresses the Production of Microbes-induced Production of IL-1β in Human Normal and Cancerous Oral Cells through the PI3K/Akt/GSK-3β Axis
p. 1, line 9:
Taiwan; hdp91111@nhri.org.tw (P.-F.H.);
--> Taiwan; chlee124@nhri.org.tw (C.-H.L.); hdp91111@nhri.org.tw (P.-F.H.);
p. 2, line 27:
outgrow
--> proliferate
p. 2, line 31:
stimulate higher levels of IL-1β
--> stimulate higher levels of IL-1β production
p. 2, line 31:
then
--> than
p. 2, lines 32-33:
Loss and gain of LDOC1 function results in increased and decreased SC5314-induced IL-1β production.
--> Loss and gain of LDOC1 function resulted in increased and decreased, respectively, CA SC5314-induced IL-1β production.
p. 2, lines 33- 34
LDOC1 deficiency resulted in increased pAktS473 upon SC5314 stimulation and inactivated pGSK-3βS9 by pAktS473.
--> LDOC1 deficiency resulted in increased active pAktS473 upon SC5314 stimulation and inactive pGSK-3βS9 phosphorylated by pAktS473.
p. 2, lines 35-36:
reduce the SC5314-stimulated IL-1β in LDOC1-deficient cells.
--> reduced the SC5314-stimulated IL-1β production in LDOC1-deficient cells.
p. 2, line 40:
outgrow
--> proliferate
p. 2, line 43:
11.3% OSCC tissues (n = 80).
--> 11.3% of OSCC tissues (n = 80).
p. 2, line 43:
an oral bacterium Fusobacterium nucleatum stimulate higher levels of IL-1β secreted by LDOC1-deficient OSCC cells then by LDOC1-expressing oral cells.
--> the oral bacterium Fusobacterium nucleatum stimulated higher levels of IL-1β secretion by LDOC1-deficient OSCC cells than by LDOC1-expressing oral cells.
p. 2, line 45:
in 4-NQO (N, a synthetic tobacco carcinogen) and arecoline (A)-cotreated mice.
--> in 4-NQO (N, a synthetic tobacco carcinogen) and arecoline (A)-cotreated mice.
Comment: It seems better not to use the abbreviations "N" and "A" in Abstract. "N" and "A" may be introduced as "NA" later (see the suggestion for the description on p, 5, lines 194-195).
p. 2, lines 46-47:
Loss and gain of LDOC1 function results in significantly increased and decreased the production of CA SC5314-induced IL-1β in oral and OSCC cell lines.
--> Loss and gain of LDOC1 function results in significantly increased and decreased, respectively, the production of CA SC5314-induced IL-1β production in oral and OSCC cell lines.
p. 2, lines 47-49:
Mechanistic studies showed that LDOC1 deficiency resulted in increased in activated Akt upon CA SC5314 stimulation and inhibitory phosphorylation of GSK-3βS9 by activated Akt.
--> Mechanistic studies showed that LDOC1 deficiency resulted in increased in activated active phosphorylated Akt upon CA SC5314 stimulation and subsequent inhibitory phosphorylation of GSK-3βS9 by activated Akt.
p. 2, lines 49-51:
PI3K and Akt inhibitors and expression of the constitutively active mutant GSK-3βS9A can significantly reduce the CA SC5314-stimulated IL-1β in LDOC1-deficient cells.
--> PI3K and Akt inhibitors and expression of the constitutively active mutant GSK-3βS9A can significantly reduced the CA SC5314-stimulated IL-1β production in LDOC1-deficient cells.
p. 2, lines 51-52:
PI3K/Akt/pGSK-3β signaling pathway
--> the PI3K/Akt/pGSK-3β signaling pathway
p. 2, line 55:
Leucine-zipper downregulation in cancer 1 (LDOC1);
--> Leucine-zipper downregulated in cancer 1 (LDOC1);
p. 2, lines 61-62:
(termed “ABC”)
--> (collectively termed “ABC”)
p. 2, line 71:
in healthy control and patients
--> in healthy control subjects and patients
p. 3, line 77:
It can become
--> It can, however, become
p. 3, line 82:
with OSCC risk and this relationship may be influenced
--> with OSCC risk, and this relationship may be influenced
p. 3, lines 84-85:
FN can induce immune suppression in gut mucosa by suppressing the functions of immune cells contributing the progression of colorectal cancer [18].
-->? FN can induce immune suppression in gut mucosa by suppressing the functions of immune cells contributing the progression of colorectal cancer[18].
p. 3, lines 86-87:
whether these microorganisms are protumor factors of POH-associated OSCC, and what may be involved in oral microbial carcinogenesis is unclear.
--> it is not clear whether these microorganisms are protumor factors of POH-associated OSCC, and what may be involved in oral microbial carcinogenesis is unclear.
p. 3, line 88:
proteomics
--> proteome
p. 3, line 98:
individuals, which promotes the development and progression of OSCC.
-->? individuals and promote the development and progression of OSCC.
p. 3, line 99:
Our previous study identified leucine-zipper, which is downregulated in cancer 1 (LDOC1), as an ...
--> Our previous study identified leucine-zipper downregulated in cancer 1 (LDOC1) as an ...
p. 3, line 101:
Analysis of LDOC1 protein expression in ...
-->? Upon analysis of LDOC1 protein expression in ...
p. 3, line 104:
we observed that high expression of LDOC1 in CGHNK2 cells,
--> we observed that high expression of LDOC1 in CGHNK2 cells,
p. 3, line 105:
whereas downregulated in DOK cell line and nearly undetectable in all OSCC cell lines tested
-->? whereas it was downregulated in the DOK cell line and nearly undetectable in all OSCC cell lines tested
p. 3, line 106:
Our previous studies demonstrated
-->? Our previous studies also demonstrated
p. 3, line 111:
similar to an SH3-binding domain
--> similar to SH3-binding domains
p. 3, line 124:
through stimulate production of oncogenic IL-1β.
--> through stimulating production of oncogenic IL-1β.
p. 3, line 126:
of CA on oral carcinogenesis in mouse with or without co-treatment of N and A.
-->? of CA on oral carcinogenesis in mouse with or without co-treatment of N and A.
Comment: 4-NQO (N) and arecoline (A) are introduced later on p. 5, lines 192-195 (see the suggestion for the description on p, 5, lines 194-195).
p. 3, line 127:
on CA and FN-induced production of IL-1β by oral normal CGHNK2 and OSCC TW2.6 cell lines
-->? on CA- and FN-induced production of IL-1β in oral normal CGHNK2 and OSCC TW2.6 cell lines
p. 4, line 131:
Candida albicans (CA)
--> CA
Comment: CA is already introduced.
p. 4. line 161:
A (green)
-->? CA(green)
p. 5, lines 166-169:
After cocultured with preparations of either live (L) or heat-killed (HK) CA SC5314 (MOI of 0.5) for 24 h, IL-1β levels in the conditioned medium of all cell tested significantly elevated, with LDOC1-deficient TW2.6 exhibited a larger increase compared with LDOC1-expressing CGHNK2 (Figure 2a).
--> After cells were cocultured with preparations of either live (L) or heat-killed (HK) CA SC5314 (MOI of 0.5) for 24 h, IL-1β levels in the conditioned medium of all cells tested significantly elevated, with LDOC1-deficient TW2.6 cells exhibiting a larger increase compared with LDOC1-expressing CGHNK2 cells (Figure 2a).
p. 5, line 174:
especially for OSCC cells with LDOC1 downregulation.
--> especially in OSCC cells with LDOC1 downregulation.
p. 5, line 176:
IL-1b in OSCC specimens
--> IL-1β in OSCC specimens
p. 5, line 179:
IL-1b mRNA expression
--> IL-1β mRNA expression
p. 5, line 180:
IL-1b in OSCC
--> IL-1β in OSCC
p. 5, line 185:
CBA analysis
--> Cytometric bead array (CBA) analysis
p. 5, line 186:
lines with or without stimulated with either live (L) or heat-killed (HK) CA SC5314 (MOI 0.5)
--> lines with or without microbes or stimulated with either live (L) or heat-killed (HK) CA SC5314 (MOI 0.5)
p. 5, line 192:
2.3. CA infection promoted the development of OSCC in 4-NQO (N) and arecoline (A)- cotreated mice.
-->? 2.3. CA infection promoted the development of OSCC in the synthetic tobacco-related carcinogen 4-NQO and the areca nut alkaloid arecoline-cotreated mice.
p. 5, lines 194-195:
, established through 4-NQO (N, a synthetic tobacco-related carcinogen) and arecoline (A, an alkaloid in the areca nut) coinduction, to assess whether the interplay between ...
--> , established through co-induction of 4-NQO and arecoline (NA), and studied whether the interplay between ...
pp. 5-6, lines 197-200:
Except for the conduction mice group without fed with NA [C0, n =5 for both mice groups with (F+) and without (F-) CA SC5314 treatment], all mice were given water containing NA for 8 (C8, n =10 for both F+ and F- mice groups) or 10 (C10, n =10 for both F+ and F- mice groups) weeks.
-->? Except for the conduction mice group without fed with NA (C0 group), all mice were given water containing NA for 8 or 10 weeks (C8 and C10 groups, respectively).
p. 6, line 201:
Starting from week 6
-->? Starting from week 8
Comment: see Figure 3a.
p. 6, line 204:
and analysis by using CHROMagar Candida plates
--> and by analysis using CHROMagar Candida plates
p. 6, line 212:
all NA-coated groups, namely C8 ... and C10
--> all NA-coated groups, namely C8 group... and C10 group
p. 6, line 217:
imply
--> implying
p. 6, lines 217-218:
that oral lesions that are repeatedly exposed to CA may be harmful
--> that oral lesions that are repeatedly exposed to CA may be harmful
p. 6, line 221:
Comments: In Figure 3a, the font used for explanation of symbols is too small in size and difficult to read. For Figure 3b, when were the samples collected? In the caption to Figure 3b, it is written that swabs were collected at week 8, 16, and 28.
p. 6, line 226:
send
--> sent
p. 6, line 228:
categories of in tongue biopsies
--> categories of in tongue biopsies
p. 7, line 231:
Comments: It would be better to explain C0, C8, C10, F-, and F+ as footnotes and to replace "SCC" by "squamous cell carcinoma".
p. 7. line 233:
oral infection of CA
--> oral infection of CA
p. 7, line 235:
(Figure 3c and Table 1)
-->? (Figure 3b, c and Table 1)
p. 7, line 242:
LDOC1 and IL-1b
--> LDOC1 and IL-1β
p. 7, line 256:
Comment: In Figure 4a, it is likely that the marks "LDOC1" and "GAPDH" are missing. In Figure 4b and c, the symbols * and ** showing p values are not shown (see p. 8, line 264).
p. 8, line 260:
(b,c) Time course CBA was used ...
--> (b,c) Time course of IL-1β production. CBA was used ...
p. 8, line 262:
withCASC5314 (MOI 0.5) (A) for indicated time.
-->? with CASC5314 (MOI 0.5) (A) for indicated time.
p. 8, line 268:
LDOC1-mediated suppressing-IL-1β function of human oral cells.
--> LDOC1-mediated suppression of IL-1β production in human oral cells.
p. 8, lines 269-270:
accounted for 2 of the top 5 kinases with enhanced phosphorylation in CGHNK2-shLDOC1 compared with CGHNK2-shCtrl (Figure 5a).
--> accounted for 2 of the top 4 kinases with enhanced phosphorylation in CGHNK2-shLDOC1 cells compared with CGHNK2-shCtrl cells (Figure 5a).
Comment: The array also contains some non-kinase phosphoprotein spots. STAT3 is a transcription factor and is not a protein kinase.
p. 8, line 275:
the LDOC1-mediated suppressing-IL-1β
--> the LDOC1-mediated suppression of IL-1β production
p. 8, line 276:
by increased inactivated pGSK-3βS9
--> by increased phosphorylated GSK-3β (pGSK-3βS9; an inactive form of GSK-3β)
p. 8, lines 277-278:
Akt can be activated by phospha-tidylinositol (3,4,5) trisphosphates (PIP3) produced by phosphoinositide 3-kinase (PI3K).
--> Akt can be activated by phosphatidylinositol 3,4,5-trisphosphate (PIP3) produced by phosphoinositide 3-kinase (PI3K).
p. 9, line 296:
represent the relative protein phosphorylation levels for the top 5 kinases exhibiting different phosphorylation states in CGHNK2-shLDOC1 and CGHNK2-shCtrl cells.
--> represent the relative protein phosphorylation levels for the top 4 kinases and the transcription factor STAT exhibiting different phosphorylation states in CGHNK2-shLDOC1 and CGHNK2-shCtrl cells (the array contained some non-kinase phosphoprotein spots).
p. 9, line 306:
Data represent
--> The bar chart represents
p. 9, line 308:
the CA-stimulated IL-1β produced by
--> the CA-stimulated IL-1β production by
p. 9, lines 310-311:
the CA-stimulated IL-1β produced by
--> the CA-stimulated IL-1β production by
p. 9, line 312:
coculture with CA SC5314 increased pAktS473 was observed in all cells
--> coculture with CA SC5314 increased pAktS473 was observed in all cells
p. 9, line 313:
with a peak more rapidly appear in
--> with a peak more rapidly appearing in
p. 9, lines 318-319:
(Figure 6d), whereas the treatment only slightly inhibited LDOC1-expressing CGHNK2-shCtrl and TW2.6-LDOC1-GFP cells.
--> (Figure 6d). whereas the treatment only slightly inhibited LDOC1-expressing CGHNK2-shCtrl and TW2.6-LDOC1-GFP cells.
Comment: The description "only slightly inhibited" would be incorrect.
p. 10, lines 320-321:
enhanced PI3K/Akt signaling caused by LDOC1 downregulation result in the augmented IL-1β stimulated by microorganisms such as CA.
--> enhanced PI3K/Akt signaling caused by LDOC1 down regulation resulted in the augmented IL-1β production stimulated by microorganisms such as CA.
p. 10, line 330:
*P< .05 versus basal activation.
Comment: In Figure 6a and b (bottom panels), asterisks are not found.
p. 10, line 347:
CA-induced IL-1β caused by LDOC1 deficiency in oral cells.
--> CA-induced IL-1β production caused by LDOC1 deficiency in oral cells.
p. 10, line 349:
TF
--> transcription factor (TF)
p. 11, line 355:
Comment: In Figure 7c, fonts are too small in size to read.
p. 11, line 359:
The bottom panels, The ratio of pGSK-3βS9A to GSK-3β protein.
--> The bottom panels shows the time course of the protein amount ratio of pGSK-3βS9A to GSK-3β protein.
p. 11, line 360:
as described in Figure 4; 3 independent experiments
--> as described in the caption to Figure 4; three independent experiments
p. 11, line 361:
were performed for IL-1β and IL-8,
--> were performed for IL-1β and IL-8,
p. 11, line 378:
that the PI3K/Akt/GSK-3β signaling pathway involved in
--> that the PI3K/Akt/GSK-3β signaling pathway is involved in
p. 12, lines 385-386:
Figure 8. Model outlines the mechanism through which LDOC1 function as an immune-modulator to attenuate the production of IL-1β in response to infection of microbes in human oral cells.
-->? Figure 8. Model outlining the mechanism through which LDOC1 functionsas an immune-modulator to attenuate the IL-1β production increased upon microbial infection of human oral cells.
p. 12, line 393:
increased the production of
--> increases the production of
p.12, line 394:
reduced the levels of
--> reduces the levels of
p. 12, line 404:
stimuli, such as growth factors, Toll-like receptors,
--> stimuli, such as those received through growth factor receptors, Toll-like receptors
p. 12, line 406:
, and PI3K/Akt/GSK-3β signaling significantly affected IL-1β production
--> , and that PI3K/Akt/GSK-3β signaling significantly affects IL-1β production
p. 12, line 407:
indicating that LDOC1 at least partly mediated immune actions
--> indicating that LDOC1 at least partly regulates immune actions
p. 12, line 410:
attachment of a small polypeptide ubiquitin that alters the property
--> attachment of the small polypeptide ubiquitin that alters the properties
p. 12, line 416:
by E3 ligase TNF receptor-associated factor (TRAF)6 when the cells are stimulated with growth factors or cytokine, which are known to induce Akt activation [42].
--> by the E3 ligase TNF receptor-associated factor (TRAF) 6 upon cellular stimulation with growth factors or cytokines inducing Akt activation [42].
pp. 12-13, line 419-420:
with E3 ligase TTC3 and MULAN
--> with the E3 ligases TTC3 and MULAN
p. 13, line 422:
We identified 2 E3 ligases
--> We identified two E3 ligases
p. 13, line 445:
k48-linked polyubiquitin chains
--> K48-linked polyubiquitin chains
p. 14, line 507:
50 nm primers
-->? 50 ng primers
p. 14, line 509:
Sample were
--> Samples were
p. 15, line 517:
week 6
-->? week 8
Comment: see Figure 3a.
p. 15, line 527:
the PBS was streak onto CHROMagar plates
--> the PBS was streaked onto CHROMagar plates
p. 15, line 534:
The production of IL-1b and IL-8
--> The production of IL-1b and IL-8
Comment: IL-8 is not determined in this work.
Reviewer 2 Report
Summary of the main findings of the study
The study by Lee and colleagues attempts to provide new information on the mechanisms of carcinogenesis of oral squamous cell carcinoma (OSCC) in a context of poor oral hygiene characterized by the exacerbated proliferation of microbes, in particular Candida albicans, leading to a local inflammatory context led through the production of IL-1β. This study explores the regulatory role of LDOC1 in the control of downstream PI3K/Akt and GSK-3β signaling pathways in the context of OSCC. In this original work the authors efficiently perform the techniques of cell and molecular biology in human cell lines as well as a mouse model of induced oral carcinoma. The methodology used is appropriate and solid, and the experimental design is clearly described. This is overall an interesting study, however the rationale underlying this study could be improved while some points need to be clarified.
Comments on the methods, results and data interpretation
Importants comments and observations
- The authors should justify the added value and the novelty provided by the data presented in Figure 1, in particular in relation to the well supplied literature on the association between the prevalence of Candida and the OSCC.
- I have concerns with the use of the TW2.6 line and the production of IL-1β. In fact, very significant differences appear between the different experiments of this study, with levels of the order of 200 pg/mL produced by the wild-type line, as shown in Figure 2b, 2,000 pg/mL produced by the TW2.6-GFP line control of TW2.6-LDOC1-GFP as presented in Figure 4c, and nearly 2,000,000 pg/mL for the TW2.6-GFP line under comparable conditions as presented in Figure 6d. This unfortunately casts doubt on the methodology employed and must be absolutely justified.
- In the conclusion, the authors should discuss in more detail the implication of pro-inflammatory effects due to infection with CA in healthy tissue and the relationship with carcinogenesis. In fact, the share of each healthy and cancerous tissue in the local production of IL-1β remains unclear, and the authors’ hypotheses regarding a possible feedback loop from the OSCC cells in the inflammatory process would be welcome and would allow the reader to better understand the choice of the authors to use an OSCC line in this study.
Minor points
- You have used the CBA kit for the detection of IL-1β and IL-8. Can you show the data with IL-8 or at least discuss it in text?
- In figure 2 the authors must use an ANOVA statistical test with corrections after multiple comparisons. A mixed-design ANOVA analysis would be beneficial, in particular to better understand the effects of live (L) and heat-killer (HK) Candida on the 2 cell lines tested. The dependent variable would then be IL-1β concentration, the within-subject identifiers would be (L) and (HK), and the between-subject identifier the cell line.
- Figures 1 and 2: can the authors discuss the capabilities of heat-killed (HK) microbes in the production of IL-1β, in particular these microbes should no longer be able to invade target cells?
- Is it possible to show GMS stainings as part of histopathology analysis shown in Figure 3d in the mouse model after infection with CA at least in C8F and C10F groups and in complement to the infection rate shown in Figure 3b?
- Figure 3b: please show the error bars on the bar plot of the infection rate in the 3 groups tested.
- Figure 4a: what is the band in the bottom line of the gel/blot?
- Figures 4b and 4c: the asterisks from the statistical test are missing, but please use a mixed-design ANOVA with the appropriate post-hoc test instead.
- Figure 5a: it appears that one of the kinase activities decreases significantly on the blot in the CGHNK2-shLDOC1 cells compared to the control cells, in particular the pair of dots located just below pair n° 4, i.e. from the 5th column (the same column as pAkt1/2/3(S473) and pGSK3α/β(S21/S0) and the 3rd line from the top. What activity is it, why not have it included in the analysis and what is the relationship with the other kinases of the pathways studied here?
- Figure 5b: it is confusing to indicate “-shCtrl” in the header of the first column on the Western blot, it would be preferable to use the label “shCtrl” as for the rest of the figure.
- Figures 5b and c: is it possible to quantify the pixels for each band under the different Western blots in order to have a more precise idea of the expression ratios? In particular the authors say that they confirm the increase in inactivated GSK-3β(S9) by Western blot (Figure 5b), which is not obvious. Indeed the pGSK-3β(S9): GSK-3β ratio could be very useful here.
- Figure 5: have the dot and Western blot experiments been replicated in another independent experiment? Please indicate it in the legend of Figure 5.
- Figure 6: it is not clear whether the calculation of Fold Change in pAkt takes into account the intensity of total Akt. Indeed on the blot in Figure 6a the intensity of the Akt bands appears to increase with the time of infection/contact with CA, and in Figure 6b to decrease again after 60 min.
- Figure 6: are the “n=2” independent experiments or duplicate of a single experiment?
- Figure 6d: the label of the y-axis is missing. The authors should consider changing the scale of the y axis or introducing a break between 1e7 and 2e7 for better visibility.
- Figure 7: as Akt works upstream of GSK-3β, I would expect to see a peak on GSK-3β activity only after the peak of Akt activity, i.e. between 90 and 120 min of co-culture with CA as shown in Figure 6a. Can you show the blot results of the quantities of active GSK-3β and inactive pGSK-3βS9A proteins for incubation times after 30 min, e.g. 60, 90, 120 and 150 min? Please also discuss the discrepancy between these timelines.
- Figure 7b: please use ANOVA for statistical testing.
- Line 535: Could you please specify the equipment and settings used for the measurement of cytokines with the CBA kit using cytometry?
- Has the data presented in Supplementary Material been published before, if not why not have included them in the Results section, in particular the whole part on the microarray analysis of CGHNK2-shLDOC1 cells discussed in the penultimate paragraph of the 'Introduction?
- Typos:
- line 31: “stimulate higher levels of IL-1β in LDOC1-deficient OSCC cells then in LDOC1-expressing oral cells” should be ‘than’
- line 44: same as above
- line 343: “…cells to secret the IL-1β after coculture with for 24 h. As…”
Round 2
Reviewer 1 Report
The manuscript was well revised; however, it is likely that newly added descriptions require revision for improvement of readability. Suggestions are shown below for author's consideration.
p. 5, lines 169-171:
IL-1β levels in the conditioned medium of all cells tested significantly elevated, with LDOC1-169 deficient TW2.6 cells exhibiting a larger increase compared with LDOC1-expressing CGHNK2 cells. Between L and HK CA, L CA can more strongly induce IL-1β than can HK CA in both cell lines tested
-->? IL-1β levels in the conditioned medium of all cells tested were significantly elevated, with LDOC1-169 deficient TW2.6 cells exhibiting a larger increase compared with LDOC1-expressing CGHNK2 cells. Between L and HK CA In both cell lines tested, L CA can more strongly induced IL-1β than can HK CA in both cell lines tested
p. 5, line 173:
after coculture for 6 hours
-->? during coculture for 6 hours
p. 6, line 229:
Representative GSM staining images showed the CA-positive samples from C8F+ and C10F+ groups.
-->? Representative GSM staining images showed of the CA-positive samples from C8F+ and C10F+ groups are shown.
p. 10, line 349-351:
After treatment with CA SC5314, pGSK-3βS9expression was observed at only 90 min after coculture with CA SC5314.
--> By treatment of CGHNK2-shLDOC1-V cells with CA SC5314, pGSK-3βS9 expression was observed at only 90 min after coculture with CA SC5314 (Figure 7a, left panel).
p. 10, lines 353-354:
In the CGHNK2-shLDOC1-GSK-3βS9A cells, the protein expression of pGSK-3βS9 decreased at 60 min and then increased significantly at 90 min, peaking at 150 min.
--> In the CGHNK2-shLDOC1-GSK-3βS9A cells stimulated with CA SC5314, the protein expression of pGSK-3βS9 decreased at 60 min and then increased significantly at 90 min, peaking at 150 min (Figure 7a, center panel).
p. 10, lines 354-356:
Ninety minutes after CA SC5314 stimulation, the expression of active GSK-3β decreased gradually, and only one-tenth of the original amount remained at 150 min in the CGHNK2-shLDOC1-V cells.
--> Ninety minutes after CA SC5314 stimulation of CGHNK2-shLDOC1-V cells, the expression of active GSK-3β decreased gradually, and only one-tenth of the original amount remained at 150 min in the CGHNK2-shLDOC1-V cells (Figure 7a, right panel).
p. 10, line 356-p. 11, line 359:
In the CGHNK2-shLDOC1-GSK-3βS9A cells, the total protein expression of GSK-3β and GSK-3βS9A, when enzyme activity was accounted for, remained stable without apparent reduction after coculture with CA SC5314.
--> In the CGHNK2-shLDOC1-GSK-3βS9A cells stimulated with CA SC5314, the total protein expression of GSK-3β and GSK-3βS9A, which accounted for the enzyme activity was accounted for, remained stable without apparent reduction after coculture with CA SC5314.
p. 11, line 374:
Comment: On lines 380-381, there is a description, *P < 0.05 and **p < 0.01 versus basal activation. In the right panel of Figure 7a, however, neither * nor ** is shown.
p. 11, lines 376-380:
(a) Western blotting analyses of the quantities of active GSK-3β, GSK-3βS9A, and inactive pGSK-3βS9 proteins in CGHNK2-shLDOC1-V and CGHNK2-377 shLDOC1-GSK-3βS9A cell lines. As presented in the right panel, densitometric analyses were performed to quantify the FC in the intensity of the active non-phosphorylated GSK-3β blots with untreated controls (time 0) set as 1.
--> (a) Western blotting analyses of the quantities of active GSK-3β, GSK-3βS9A, and inactive pGSK-3βS9 proteins in CGHNK2-shLDOC1-V (left panel) and CGHNK2-377 shLDOC1-GSK-3βS9A (center panel) cell lines. As presented in the right panel, Densitometric analyses were performed to quantify the FC in the intensity of the active non-phosphorylated GSK-3β blots with untreated controls (time 0) set as 1 (right panel).
p. 12, line 394:
have yet to investigate cases from Taiwan.
-->? have yet require to further investigate cases from Taiwan.
p. 17, lines 624-626:
Accordingly, to prevent OSCC, we recommend smoking cessation to prevent LDOC1 silencing, which can cause immune abnormalities, and good dental hygiene to prevent the proliferation of microbes, such as CA or FN, in oral cavities.
--> Accordingly, to prevent OSCC, we recommend smoking cessation to prevent avoid LDOC1 silencing, which can cause immune abnormalities, and good dental hygiene to prevent not to allow the proliferation of microbes, such as CA or FN, in oral cavities.
Author Response
REVIEWER 1
We thank you for your careful editorial review and suggestions. We have corrected the typological and grammatical errors that you identified on pages 5, 6, 10, 11, 12, and 17. We respond to your comment on Figure 7a as follows.
Comment 1
On lines 380-381, there is a description, *P < 0.05 and **P < 0.01 versus basal activation. In the right panel of Figure 7a, however, neither * nor ** is shown.
Answer 1
Thank you for your comments. We have modified Figure 7a by adding asterisks to indicate a statistically significant increase and decrease in the protein amount of active GSK-3β. The revised Figure 7a (page 11, line 377):

Reviewer 2 Report
The Reviewer thanked the authors for taking into consideration its recommendations answering all the questions and providing some additional data, which allowed to take away any doubt and correct the last minor errors. The interest of this study was not to be questioned, the work provided is remarkable, the experiments were made in the best way and the analyzes as well as the preparation of the manuscript were very well done. I have no further point to add.
Author Response
We would like to thank the reviewer for reading our manuscript and reviewing it. The reviewer's comments were very insightful and enabled us to improve the quality of our manuscript entitled "LDOC1 suppresses microbe-induced production of IL-1β in human normal and cancerous oral cells through the PI3K/Akt/GSK-3β axis".